# Accidental Encounter of Repair Intermediates in Alkylated DNA May Lead to Double-Strand Breaks in Resting Cells

**DOI:** 10.3390/ijms25158192

**Published:** 2024-07-26

**Authors:** Shingo Fujii, Robert P. Fuchs

**Affiliations:** 1Cancer Research Center of Marseille, Department of Genome Integrity, CNRS UMR7258, Inserm U1068, Institut Paoli-Calmettes, Aix Marseille University, 13273 Marseille, France; 2SAS bioHalosis, Zone Luminy Biotech, 13009 Marseille, France

**Keywords:** chemotherapy, DNA damages, DNA repairs, DNA double-strand breaks, repair accident model

## Abstract

In clinics, chemotherapy is often combined with surgery and radiation to increase the chances of curing cancers. In the case of glioblastoma (GBM), patients are treated with a combination of radiotherapy and TMZ over several weeks. Despite its common use, the mechanism of action of the alkylating agent TMZ has not been well understood when it comes to its cytotoxic effects in tumor cells that are mostly non-dividing. The cellular response to alkylating DNA damage is operated by an intricate protein network involving multiple DNA repair pathways and numerous checkpoint proteins that are dependent on the type of DNA lesion, the cell type, and the cellular proliferation state. Among the various alkylating damages, researchers have placed a special on O^6^-methylguanine (O^6^-mG). Indeed, this lesion is efficiently removed via direct reversal by O^6^-methylguanine-DNA methyltransferase (MGMT). As the level of MGMT expression was found to be directly correlated with TMZ efficiency, O^6^-mG was identified as the critical lesion for TMZ mode of action. Initially, the mode of action of TMZ was proposed as follows: when left on the genome, O^6^-mG lesions form O6-mG: T mispairs during replication as T is preferentially mis-inserted across O^6^-mG. These O6-mG: T mispairs are recognized and tentatively repaired by a post-replicative mismatched DNA correction system (i.e., the MMR system). There are two models (futile cycle and direct signaling models) to account for the cytotoxic effects of the O^6^-mG lesions, both depending upon the functional MMR system in replicating cells. Alternatively, to explain the cytotoxic effects of alkylating agents in non-replicating cells, we have proposed a “repair accident model” whose molecular mechanism is dependent upon crosstalk between the MMR and the base excision repair (BER) systems. The accidental encounter between these two repair systems will cause the formation of cytotoxic DNA double-strand breaks (DSBs). In this review, we summarize these non-exclusive models to explain the cytotoxic effects of alkylating agents and discuss potential strategies to improve the clinical use of alkylating agents.

## 1. Introduction

Chemotherapy is an indispensable approach to tackling a variety of diseases in hospitals [1,2]. Whether a chemical during chemotherapy is effective largely depends upon the cellular response that deals with the chemical. The representative chemical during the chemotherapy is generally an alkylating agent that leads to DNA damage [2,3]. Human cells have a variety of defense mechanisms to fix DNA damage using intrinsic DNA repair systems. DNA damages induced by alkylating agents are usually repaired in three distinct categories of DNA repair pathways: direct reversal repair, base excision repair (BER), and mismatch repair (MMR). For instance, the direct reversal repair system includes direct conversion of the O^6^-methylguaine (O^6^-mG) lesion to guanine through the suicidal enzymatic reaction of O^6^-methylguanine-DNA methyltransferase (MGMT), direct conversion of 1-methyladenine (1-mA) and 3-methylcytosine (3-mC) lesions to the adenine and cytosine, respectively, through the enzymatic reaction of alkylated DNA repair protein B homolog (ALKBH) proteins. While repair via direct reversal is obviously the best way to maintain genetic integrity, only a small number of DNA lesions are repaired by direct reversal. BER pathways are responsible for the repair of a variety of additional lesions, such as N3-methyladenine (N3-mA) and N7-methylguanine (N7-mG) [4,5,6]. If O^6^-mG lesions escape direct reversal by MGMT, during replication they are efficiently converted into pre-mutagenic O^6^-mG: T mispairs that were shown to be recognized by the post-replication mismatch DNA repair pathway (MMR) [7,8]. Thymine opposite the O^6^-mG lesion is recognized as a DNA replication error in the context of MMR because the thymine is located on the nascent strand. During the process of MMR following the T-containing strand removal, T is frequently re-inserted across template O^6^-mG triggering thus reforming the initial O^6^-mG: T mispair and triggering a novel MMR attempt, leading to the so-called “futile MMR cycle”. These futile MMR cycling events have tentatively been proposed to ultimately lead to DSBs [9] (Figure 1). However, the molecular mechanisms supporting the formation of DSBs have remained elusive [10]. In contrast to detailed knowledge of the molecular mechanisms operating for individual DNA repair pathways, relatively little is known regarding crosstalks between repair pathways [6]. In order to obtain a more integrated view of potential interference between DNA repair pathways that operate simultaneously during exposure of DNA to alkylating agents, we implemented biochemical assays in Xenopus egg extracts [11]. Based on our experimental results, we proposed a model during which accidental interaction between BER and MMR pathways, simultaneously acting at closely spaced lesions, leads to DSBs [11,12]. In addition to the futile cycle model, the model that we propose adds a further dimension to the debate as it provides a molecular mechanism of O6-mG-induced cytotoxicity and cell death in the case of non-dividing cells that represent the vast majority of cells in our bodies. In this review article, we will describe how the accidental encounter of two independent repair events, MMR and BER, taking place simultaneously at closely-spaced alkylation adducts in opposite DNA strands can lead to a DSB. In addition, we are suggesting a similar model for the cytotoxic effects of methylazoxymethanol (MAM), a hydrazine-related chemical, with carcinogenic and neurotoxic potential [13,14,15].

## 2. Clinical Use of Temozolomide (TMZ)

Alkylating DNA damages in genomic DNA results from cellular metabolic products (endogenous alkylation) and exogenous chemicals such as nitroso-compounds and chemotherapeutic agents [4,6]. With respect to chemotherapy, the alkylating agent temozolomide (TMZ) is widely used for the treatment of glioblastomas [1,2]. TMZ belongs to the group of triazene compounds and is the predominant mono-functional DNA alkylating agent used in the treatment of glioblastoma in combination with surgery and ionizing radiations [2,16,17]. During chemotherapy, following oral administration, TMZ is rapidly converted into the active metabolite MTIC [5-(3-methyl-1-triazeno) imidazole-4-carboxamide] by cellular metabolic processes. After that, the MTIC spontaneously decomposes into a methyldiazonium ion which directly reacts with genomic DNA at the N7 position of guanine (N7-mG), the N3 position of adenine (N3-mA), and the O^6^ position of guanine (O^6^-mG) as well as minor adducts at the N1 position of adenine (1-mA) and the N3 position of cytosine (3-mC). In order to repair the diversity of alkylating DNA damages, human cells utilize at least three DNA repair processes, the BER system, the MMR system, and direct reversal repair proteins such as MGMT [18] and the ALKBH proteins [5,6]. Intriguingly, whereas the O^6^-mG lesion represents a minor fraction (i.e., <10%) among all TMZ-induced alkylating DNA lesions, these lesions turn out to be the most cytotoxic and mutagenic. Related to the cytotoxic effects, it is observed that DSBs trigger apoptosis after TMZ treatment [19]. The significant adverse effects of the O^6^-mG lesion were demonstrated through studies involving the direct reversal repair protein MGMT. When MGMT is highly expressed in cells, the alkylating agent-induced cell death phenomenon is largely blocked, thus highlighting the cytotoxic effect of O^6^-mG lesions. In contrast, upon loss of MGMT expression, cells become highly sensitive to alkylating agents [10,20]. These studies indicated that the presence of the O^6^-mG lesion on genomic DNA is a critical factor for TMZ-mediated cancer therapy. Therefore, different approaches aimed at maintaining O^6^-mG lesions on genomic DNA, such as depleting or inhibiting the MGMT protein, have been developed [20]. As an alternative mechanism leading to TMZ resistance, it has been reported that efficient homologous recombination (HR) pathway can rescue glioblastoma-derived tumor cells from TMZ-mediated cytotoxic effects [21,22].

## 3. The MMR-Mediated Futile Cycle Model Driven by the Persistence of O^6^-mG Lesions in the Template Strand

Based upon the critical observation that O^6^-mG is the lesion that is responsible for cell death, led to the so-called futile (or abortive) MMR cycle model [8,23] (Figure 1). When O^6^-mG lesions escape direct MGMT-mediated reversal, the replicative DNA polymerase predominantly incorporates thymine and, to a lesser extent, cytosine opposite the O^6^-mG lesion on the template DNA during the DNA replication. The resulting base-pairing geometry remains essentially unperturbed, and the T: O^6^-mG base pair evades the proofreading function associated with the replicative DNA polymerase. Frequent incorporation of T opposite the O^6^-mG lesion turns out to be mutagenic, inducing GC to AT transitions [24]. Since O^6^-mG lesions do not prevent the progression of DNA replication, the recurrent occurrence of O^6^-mG: T mispairs leads to MMR-mediated cytotoxicity via a so-called “futile cycle model” as follows. As the replicative DNA polymerase re-incorporates thymine opposite the O^6^-mG lesion thus re-forming the O^6^-mG: T mispair that is recognized as a DNA replication error in the context of the post-replication mismatch DNA repair pathway. The thymine on the nascent strand in the O^6^-mG: T mispair is recognized by the MutSα complex composed of MSH2 and MSH6 proteins as a replication error. The MutSα complex in turn recruits the MutLα complex composed of MLH1 and PMS2 proteins. After that, an endonuclease activity associated with the MutLα complex introduces a nick(s) in the T-containing nascent strand leading to the formation of a large single-stranded DNA gap containing a persistent O^6^-mG lesion. During MMR synthesis, thymine is frequently re-incorporated opposite the O^6^-mG lesion [24], thus forming again the O^6^-mG: T mismatch base pair, thus the concept of abortive or futile mismatch repair cycles [8,25] as proposed over 40 years ago [23]. The intrinsic property of MMR to remove the nascent strand inevitably leads to the permanence of the O^6^-mG lesion in the parental strand. This situation is quite unique to O^6^-mG when compared to base analogs such as 2-AP or BrdU that get incorporated in the nascent strand forming mismatched which are efficiently removed by the MMR system. Within the framework of the futile cycle model, with respect to cytotoxicity, it is supposed that the repeated rounds of excision and re-synthesis will eventually lead to activation of the ATR/CHK1 signaling cascade and the onset of apoptosis although no precise mechanism is available yet [4,6,26,27].

## 4. The MMR-Mediated Direct Signaling Model Complementing the Futile Cycle Model

There is a complementary model termed the “direct signaling model” that has been proposed to explain the resulting onset of apoptosis induced by the presence of the O^6^-mG lesion dependent upon the DNA replication. It is suggested that recognition of the O^6^-mG: T mispair by MMR complexes MutSα and MutLα, further elicits the recruitment of the DNA damage response proteins ATR, ATRIP, and TopBP1 resulting in activation of the DNA damage checkpoint response (DDR) [28,29]. In that model, the MutSα complex directly interacts with ATR, TopBP1, and Chk1 while the MutLα complex interacts with TopBP1 [30]. In conclusion, the DDR checkpoint signal cascade is activated via the activation of the MMR system, triggering the onset of cell cycle arrest and apoptosis. Both models (“futile cycle” and “direct signaling”) seem to fit the observation that normal stem cells exhibit MMR-dependent DDR signaling in response to O^6^-mG: T mismatches within the first S phase [31]. Overall, both O^6^-mG lesion-induced cell death models are in good agreement with published features, namely that a functional MMR pathway is indispensable [9], that impaired MGMT is essential [18], and that the occurrence of DNA strand breaks is positively correlated [10]. However, neither of these models can explain why the MMR-dependent DDR response requires two rounds of DNA replication in cancer cells [31].

## 5. The DNA Repair Accident Model to Explain the Cytotoxic Effects of the Alkylating Agent

In combination with surgery and radiation during the clinical treatment of glioblastoma, the alkylating agent TMZ is used as a chemotherapeutic agent. Despite its many years of clinical use, the mode of cytotoxic action of TMZ remains elusive. Like other alkylating agents, TMZ induces a broad spectrum of DNA lesions, including N7-mG (70–75%), N3-mA (8–12%), and O^6^-mG (8–9%). As described above, distinct DNA repair systems deal with these lesions, namely BER for N7-mG and N3-mA lesions, direct reversal repair via MGMT protein, or the MMR system for the O^6^-mG lesion. As discussed above, the cytotoxic mode-of-action of alkylating agents such as TMZ is believed to result from MMR-mediated “futile cycling” and/or “direct signaling” at O^6^-mG: T mispairs. Furthermore, in addition to MMR, BER-derived repair intermediates have been proposed to contribute to the cytotoxic effects of TMZ [32,33,34]. Alternative models involving crosstalk between the BER and the MMR systems [35] or accumulation of BER repair intermediates [32] suggest the occurrence of DNA double-strand breaks (DSBs) to explain cytotoxicity.

In contrast to the aforementioned models (“futile cycling” and “direct signaling”) both of which require replication to form O^6^-mG: T mismatches (S phase), the vast majority of cells in glioblastoma tumors are non-dividing or quiescent [36]. Therefore, investigating the mechanism of action of TMZ in the absence of replication is of critical importance. To this end, we analyzed the fate of TMZ-damaged DNA in extracts (Xenopus egg extracts) in the absence of replication. For that purpose, a recently developed plasmid DNA pulldown approach, termed IDAP (which stands for the isolation of DNA-associated proteins), was implemented. This approach, which aims at capturing nucleoprotein complexes formed on specific plasmids in nuclear extracts, was shown to be highly efficient and versatile [37,38,39]. Briefly, the core aspect of this methodology is to immobilize, on magnetic beads, a circular plasmid containing the DNA of interest. Plasmid immobilization is mediated by means of a specific oligonucleotide able to form a triple helix (TFO probe) with a cognate double-stranded DNA sequence present on the plasmid; the other extremity of the TFO probe harbors a biotin moiety that interacts with streptavidin-conjugated magnetic beads. To implement this approach in the case of alkylation lesions, plasmid DNA treated by N-methyl-N-nitrosourea (MNU: a TMZ mimic) was incubated with cell-free extracts prepared from Xenopus laevis eggs under non-replicating conditions. In this methodology, many proteins were isolated irrespective of the presence or the absence of DNA damage. Following the identification of the captured proteins by means of mass spectrometry (MS) analysis, a variety of proteins were specifically recruited by the presence of MNU-induced DNA damage compared to the undamaged plasmid. Interestingly, most MMR core proteins were highly enriched despite the absence of DNA replication. Biochemical assays were subsequently implemented to validate the MS data. It was revealed that proteins of both the MMR and the BER systems are active on the damaged DNA treated by MNU. From the biochemical data, we concluded that when MMR and BER repair processes operate independently on lesions located in the same DNA molecule in opposite strands, a double-strand break may result from the accidental encounter of these two repair intermediates [11]. We refer to such an event as a “repair accident” (Figure 2) [12].

## 6. Genotoxic Impact of Methylazoxymethanol (MAM)

MAM, which is a hydrazine-related chemical, is naturally found in foods including mushrooms and plants, possessing carcinogenic and neurotoxic potential (i.e., they are linked to cancer and neurological diseases) once they are metabolized in the liver [13,14,15]. Like TMZ, the activated MAM induces a variety of DNA damages (e.g., O^6^-mG, N7-mG, 8-oxo-G), resulting in the activation of multiple DNA repair systems including the MMR and the BER systems. Whereas the MAM-treated postmitotic cells undergo apoptosis or non-apoptotic cell death, the mechanism of action remains elusive. Given the similarity between MAM and TMZ, in adduct formation, we suggest that cytotoxic effects of MAM in the postmitotic cells might be achieved within the framework of the “repair accident model” described above.

## 7. Discussion

From numerous studies in numerous cellular and animal models, it has been concluded that the cytotoxicity of alkylating agents, including the most crucial therapeutic agent TMZ, is mainly due to O^6^-mG lesions. The cytotoxic cascade depends on both MMR and DNA replication. To explain these features, two models (futile cycle and direct signaling models) have been proposed. In these models, the target of the MMR system is the O^6^-mG: T mispair formed during the DNA replication [8]. However, since these models cannot explain all of the experimental observations, it suggests that there may be a missing piece to fully explain the alkylating agent-induced cytotoxic effects. In particular, the proposed models require DNA replication, a feature that is not encountered in most glioblastoma tumor cells that are non-dividing or quiescent [36]. In order to explain the intricate phenomena involved in alkylating agent-induced cytotoxic effects, a crosstalk between the MMR and the BER systems has been suggested [33,34,35]. The “repair accident model” proposed here suggests a mechanism for the formation of cytotoxic DSBs in the absence of DNA replication by virtue of an accidental encounter of MMR and BER repair intermediates.

When considering the clinical use of alkylating agents based on the repair accident model (Figure 2), as DSBs are formed as a consequence of concomitant processing of lesions by proteins of both the MMR and BER systems, enhancement of the cytotoxic effects may be achieved by partially impairing these repair systems, namely by slowing down the latter events such repair synthesis or ligation steps in either BER and/or MMR.

In addition to the pre-existing two models (futile cycle and direct signaling models), the repair accident model will become the third model to compensate for the weak points of the pre-existing two models that are incompatible with the cytotoxic effects in non-dividing or quiescent cells. Furthermore, the idea of DSB formation in the repair accident model may become a molecular basis to explain the cytotoxic effects of MAM-inducing cancer and neurological disease in non-dividing or quiescent cells.

## 8. Patents

R.P.F. and S.F. hold a patent, 100007204, which covers the conceptualization and methodology described in this manuscript.

## Figures and Tables

**Figure 1 ijms-25-08192-f001:**
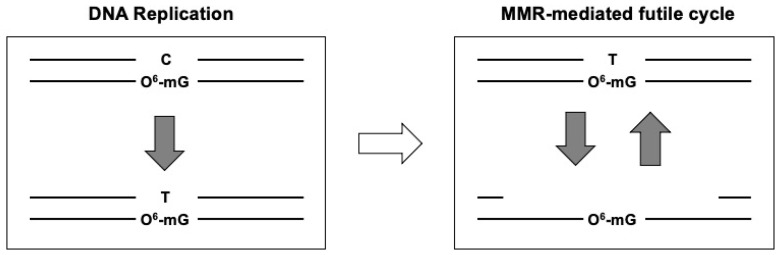
The MMR-mediated futile cycle model is driven by the persistent presence of the O^6^-mG lesion on the genomic DNA. Alkylating agents such as temozolomide (TMZ). Widely used in the clinic to treat glioblastomas, alkylating agents induce a broad spectrum of adducts (lesions) on genomic DNA. In the MMR-mediated futile cycle model, the O^6^-mG lesion among alkylating agent-induced DNA damages is exclusively focused as the only source related to alkylating agent-induced cell death phenomenon. During DNA replication, thymine is preferentially incorporated opposite the O^6^-mG, forming the O^6^-mG: T mispair (left panel). This mismatch activates the MMR system. However, the O^6^-mG lesion remains persistently present in the parental strand, resulting in a novel MMR attempt that is futile by nature (right panel). Such iterative rounds of the MMR repair process could ultimately lead to cell cycle arrest and apoptosis.

**Figure 2 ijms-25-08192-f002:**
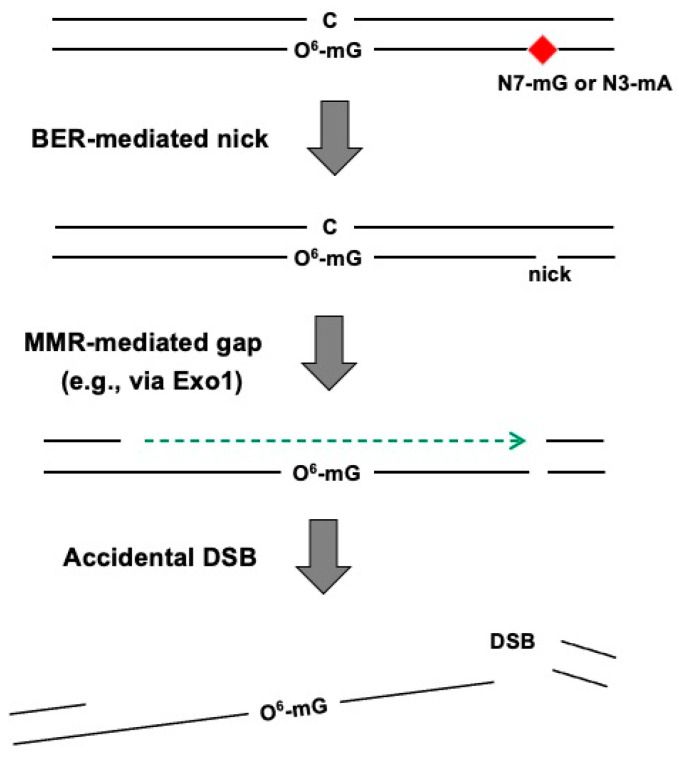
The Repair Accident Model. In the case of treatment with an alkylating agent such as TMZ during chemotherapy, a variety of DNA damages appear and are repaired by multiple DNA repair pathways. Indeed, distinct DNA repair systems work independently depending upon their substrate specificities. The BER system mainly acts at the N7-mG or the N3-mA lesion, while the core MMR proteins recognize not only the O^6^-mG: T base pair but also the O^6^-mG: C base pair as shown by our work [11]. Therefore, even in non-dividing or quiescent cells treated by alkylating agents; when an N7-mG or an N3-mA lesion is closely located with the O^6^-mG lesion (e.g., within several hundred nucleotides), the accidental encounter of BER and MMR derived repair intermediates were shown to lead to DSB when they occur within the same time frame. In non-dividing cells treated by an alkylating agent, the “repair accident model” scenario is as follows: when the MMR system recognizes the O^6^-mG: C base pair, the mechanism of initiation of the MMR reaction is presently unknown. It is likely that the strand discrimination signal is provided by a BER-mediated nick equally likely to occur in either strand. This is in contrast to the situation that occurs during replication where MMR is directed toward the excision of the nascent strand. In any case, Exo1-mediated strand degradation or helicase unwinding assists the DNA gap formation. If, within the same timeframe, the MMR-mediated gap formation process encounters, in the opposite strand, a nick resulting from an intermediate, independent BER repair, a DSB will result. Red square indicates a DNA lesion that is repaired by the BER system. Green arrow shows degradation of a strand by an exonuclease.

## Data Availability

Not applicable.

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
