# Peer review of "Accidental Encounter of Repair Intermediates in Alkylated DNA May Lead to Double-Strand Breaks in Resting Cells"

_ijms, 2024, doi:10.3390/ijms25158192_

Round 1
Reviewer 1 Report
Comments and Suggestions for Authors
This is an excellent manuscript that will be broadly appreciated. It is well written and it lays out important information about interpathway interactions and their importance in response to methylation damage. Suggestions follow.
At the start, futile cycling is described very briefly. It would be good to say that the mechanism is described in more detail below since you don't really explain the reason for MMR taking out T instead of O6meG.
What exactly does MAM do? Does it create a methyldiazonium ion? Also, there are at least two other chemo agents that create methylated bases.
Overall, what do we know about the ability of MMR to act outside of S phase? One thing you could report is that MMR is active when people to CRISPR, and that most cells are not in S phase during CRISPR.
Was Marinus the first to put forward the futile cycling model? If so, that reference should be at line 115.
Overall, it is not made sufficiently clear why MMR takes out T and not O6meG. A more thorough description including teaching the reader that MMR always removes the newly synthesized strand because it assumes that it was a polymerase misinsertion.
Are you assuming that this entire story depends on lesions encountered by the leading strand? What happens to the lagging strand. Some ideas there would be helpful.
Since a big part of the story is how the new xenopus method is used, it would be great if there was a figure explaining.
Conspicuously absent is a figure showing a fork encountering a MMR-driven gap.
Figure 2: is the nick on the left of O6MeG? If so, which exo goes 5' to 3'? Also, the dotted blue line indicates replication, but then the replicated DNA is missing from the bottom part of the figure.
Also, wouldn't the nick from BER serve to direct MMR to take out the same strand? We know that nicks direct MMR.
What is known about toxicity in non-dividing cells? Do we know that non-dividing cells respond to TMZ in glioblastomas? What about other cell types? It seems that most literature points to non-dividing cells as being resistant to TMZ.
The point about neurons is interesting, but couldn't it still be direct signaling? or MMR-gap driven signaling that is independent of replication?
Minor:
Line 49 should read repaired rather than concerned.
Line 124: After "cycle model" add "as follows".
Line 154 needs references
Reviewer 2 Report
Comments and Suggestions for Authors
The authors present a review article highlighting the interactions between DNA repair pathways, suggesting that simultaneous repair activities at closely spaced lesions can lead to DNA DSBs, contributing to cytotoxicity and cell death. This repair accident model proposes that accidental encounters between BER and MMR intermediates cause DNA DSBs even in non-dividing cells. This is crucial for understanding TMZ's cytotoxic effects in quiescent cells. This model offers a fresh perspective that addresses limitations in existing models, especially for non-dividing cells. Experiments will be needed to validate the model and know the molecular mechanisms underlying the formation of these DNA DSBs.
Minor comment:
-There are three important observations that the authors should include in chapter 2. First, DNA DSBs as the lesions triggering apoptosis downstream after TMZ treatment (DOI: 10.1038/sj.onc.1209785). Second, there are other mechanisms that could explain TMZ resistance in addition to MGMT: MGMT-deficient orthotopic GBM-derived tumor cells that are significantly resistant to TMZ do not exhibit MGMT upregulation or loss of MMR activity but rather a more accelerated repair of TMZ-induced DSBs through a more efficient Homologous Recombination (HR) pathway (DOI: 10.1158/1541-7786.MCR-16-0125). Finally, within GBM tumors, there are glioma stem-like cells, a cellular subset capable of significantly expanding and generating new tumors, being highly resistant to TMZ due to a higher HR repair activity (DOI: 10.1007/s12035-022-02915-2).
Comments on the Quality of English LanguageI recommend the authors to polish the redaction a bit more using the free software Grammarly
